# Teacher Preparedness for Medical Emergencies in Belgian Classrooms: Studying Objective and Subjective First-Aid Knowledge

**DOI:** 10.3390/children10040669

**Published:** 2023-03-31

**Authors:** Maya Vermonden, Liesse Dehaerne, Jaan Toelen, David De Coninck

**Affiliations:** 1Faculty of Medicine, KU Leuven, 3000 Leuven, Belgium; maya.vermonden@student.kuleuven.be (M.V.); liesse.dehaerne@student.kuleuven.be (L.D.); 2Leuven Child and Youth Institute, KU Leuven, 3000 Leuven, Belgium; jaan.toelen@uzleuven.be; 3Department of Development and Regeneration, KU Leuven, 3000 Leuven, Belgium; 4Department of Pediatrics, University Hospitals Leuven, 3000 Leuven, Belgium; 5Centre for Sociological Research, KU Leuven, 3000 Leuven, Belgium; 6Institute of Educational Science, University of Graz, 8010 Graz, Austria

**Keywords:** first aid, teachers, Belgium, surveys and questionnaires

## Abstract

About one in seven accidents involving children occurs at school. Roughly 70% of these accidents involve children under the age of 12 years. Thus, primary school teachers may be confronted with accidents where first aid could improve the outcome. Despite the importance of first-aid knowledge among teachers, little is known about this topic. To fill this gap, we conducted case-based survey research on the objective and subjective first-aid knowledge of primary school and kindergarten teachers in Flanders, Belgium. An online survey was distributed to primary school and kindergarten teachers. It included 14 hypothetical first-aid scenarios in a primary school context to assess objective knowledge, along with one item to assess subjective knowledge. A total of 361 primary school and kindergarten teachers completed the questionnaire. The participants achieved an average knowledge score of 66%. Those who had completed a first-aid course had significantly higher scores. Knowledge regarding child CPR was amongst the lowest, with only 40% correct answers. Structural equation modeling showed that only previous first-aid training, recent experience with first aid, and subjective first-aid knowledge were linked to teachers’ objective first-aid knowledge—particularly for basic first aid. This study shows that completing a first-aid course and completing a refresher course can predict objective first-aid knowledge. We therefore recommend that first-aid training and regular refresher courses should be mandatory in teacher training, given that a large share of teachers may need to apply first aid to a pupil at some point during their career.

## 1. Introduction

First aid, as described by the European Resuscitation Council (ERC), is the initial care provided for an acute illness or injury [1]. There are multiple goals of first aid, including preventing death or serious disability, preventing further injury or illness, and facilitating recovery. Some key principles of the provision of first aid include recognizing, assessing, and prioritizing the need for first-aid care. This has to be done using appropriate competencies based on the most optimal available evidence. Another important characteristic of first aid is recognizing its limitations and seeking additional care, such as activating emergency medical services or other medical assistance. A unique characteristic of first aid—compared to other medical interventions—is its universal nature. It can be learned and applied by anyone in any situation that requires first aid, including self-care [1,2].

One of the main objectives of the ERC is to ensure that every European knows and properly applies basic cardiopulmonary resuscitation (CPR) [1]. In Flanders—the northern region of Belgium and the setting of the present study—large-scale efforts are being made to disseminate knowledge about first-aid procedures. For example, more attention is being paid to first-aid classes for students at school. The attainment targets for first aid in the first and second grades of secondary education have just been renewed, and the attainment targets for the third grade will be renewed by September 2023 [3,4,5].

Children are more likely than adults to suffer minor and major trauma (e.g., falling from heights, burns, abrasions, etc.), because they are regularly exposed to new experiences during their physical and social development in childhood and adolescence, are less able to assess risks, and are more playful [2,6,7]. Given that children spend a considerable amount of time at school during these formative years, it does not come as a surprise that accidents that require first aid take place at schools on a daily basis.

In Belgium, about one in seven accidents involving children occurs in the school context. Furthermore, roughly 70% of these accidents involve children under 12 years of age. Fractures and bruises (27%), injuries to the head and brain (21%), and burns (14%) are the most common types of school accidents [8]. In 2020, a total of 727,167 pupils were enrolled in primary education in Flanders (age range: 6 to 12 years) [9]. Primary school teachers, who are responsible for these pupils when they are at school, are thus regularly confronted with situations in which first aid must be applied [7].

Despite the fact that teachers are confronted with various situations in which first aid can and should be applied, teacher training institutes in Flanders are currently not obliged to include a standardized first-aid course in their curricula. The majority of teacher training colleges offer voluntary first-aid courses, but there is no quality control of the contents of these courses [10,11,12,13]. In addition, studies have shown that a one-time first-aid course during teacher training is insufficient to maintain an adequate level of first-aid skills throughout the teaching career, making refresher courses necessary [14]. When looking at the Belgian population specifically, this is confirmed once again. A large-scale first-aid survey conducted by the Belgian Red Cross in 2017 showed that 80% of participants believed that they would know what to do in various emergency situations. However, the study also showed that 35% of people overestimated their knowledge of first aid [15].

When focusing on first-aid knowledge among primary school teachers in Flanders, no studies have been conducted on this topic to the best of our knowledge. Studies investigating teachers’ first-aid knowledge in other countries have shown that it is often insufficient [2,7,16,17,18,19,20,21,22,23,24,25], and first-aid training is paramount to improve this knowledge [26,27]. The aim of this study is to map the objective and subjective first-aid knowledge among teachers in primary education (kindergarten and primary education) in Flanders, Belgium, and to study which individual characteristics are linked with the subjective and objective knowledge of first aid. Regarding these two types of knowledge, we follow the definition outlined by Han [28]: “Objective knowledge refers to how much an individual actually knows and subjective knowledge (also known as perceived or self-assessed knowledge) involves how much an individual thinks he/she knows […]”.

## 2. Materials and Methods

### 2.1. Study Design

We conducted an online survey between March and July 2022. To assess objective first-aid knowledge, we developed and presented 14 hypothetical first-aid scenarios in a primary school context. After the survey was designed, it was programmed into the digital survey tool Qualtrics. Because of the positive effect of peer review on the quality of multiple-choice examinations [29], the survey was piloted in 13 individuals, including both medical experts (A&E physicians) and medically untrained individuals (content and language comprehension). Adjustments to the survey (mainly to the phrasing of the scenarios) were made based on this feedback.

Before starting the survey, the participants were informed about the nature and purpose of the study, the anonymity of the survey, and that their participation was entirely voluntary. Contact details of the researchers for any further questions were also provided. This study was approved by the ethical committee of the host university (case number MP018810).

The only inclusion criterion was that all participants had to be active teachers in kindergarten (2.5–6-year-old children) or primary education (6–12-year-old children) in the Flemish school system (total population size: 82,520 teachers) [9]. Interns and special education teachers were not included in the study. All teachers had to have had their teaching training in Flanders.

This study was conducted in primary schools and kindergartens in Flanders. For the distribution of our survey, we utilized a snowball sampling technique through digital media. We initiated our data collection by contacting schools through emails to principals. We also distributed the survey through a closed Facebook group for primary school teachers. This resulted in 361 valid survey responses.

The 14 scenarios designed to assess objective first-aid knowledge consisted of multiple-choice questions (see Appendix A). Based on the severity of the cases, we categorized these items into two categories: life-threatening (e.g., CPR, …) and basic first aid (e.g., dental injury, …) cases. We calculated knowledge indices that indicated the extent to which respondents provided correct answers to the items (0 = no correct answers, 1 = all items answered correctly). These were calculated for all 14 items, but also for the life-threatening and basic first aid items separately. 

Subjective first-aid knowledge was measured through a single item asking respondents to rate their first-aid knowledge (1 = very good, 5 = very poor; reverse-coded prior to analysis).

We also collected data on a number of first-aid characteristics of the participants, i.e., whether they had ever completed a first-aid course, either during their teacher training or on their own initiative (yes/no); whether those who had completed a first-aid course had also completed at least one refresher course (yes/no); whether the school at which they worked had a so-called “first aid manager” (i.e., a staff member who is the main point of contact when first aid is required); and whether the participants had needed to use first aid at school over the past three years (yes/no). Additionally, we also asked about the participants’ gender and years of experience in teaching.

### 2.2. Statistical Analysis

We used independent-samples t-tests to determine whether there were statistically significant differences with regards to overall, life-threatening, and basic first-aid knowledge, as well as subjective first-aid knowledge, between (a) teachers with or without first-aid training and (b) teachers who had or had not completed refresher courses. Subsequently, we designed a structural equation model to assess the links between first-aid indicators (e.g., training, recent use, manager at school), subjective first-aid knowledge, and objective knowledge of life-threatening and basic first-aid cases, as presented in Figure 1. We also controlled for the participants’ gender and years of teaching experience.

## 3. Results

Most of the respondents (95%) were female. Although the respondents were mostly recruited through social media, over 50% of the participants had over 10 years of teaching experience (60%), and about one-third (34%) of the respondents had been primary school teachers for over 20 years. In total, 80% of participants reported that they had taken a first-aid course at some point, with 38% of this group additionally taking one or more refresher courses. Additionally, 66% of the respondents had applied first aid at school in the past 3 years. The participants achieved an average overall objective knowledge score of 66% on the 14 case-based first-aid questions, with an average of 82% on the questions in the category of life-threatening cases and 53% in the basic first aid category (Table 1).

An overview of the participants’ (correct) responses to the cases can be found below. In Table 2, we also highlight which cases were classified as life-threatening and which were classified as basic first aid. For the specific wording of each case, see Appendix A While we observed that nearly all respondents knew what to do in most life-threatening situations, we also noted that for both cases in which CPR needed to be applied—to an adult and to a child—a lower share of respondents were able to provide the correct course of action. Knowledge of CPR on children was particularly lacking, with only 40% of respondents correctly knowing what to do in such situations, which could lead to critically dangerous situations in real-life settings.

Table 3 presents the results of independent-samples t-tests on objective and subjective knowledge scores by respondents who did (not) have first-aid training; and by those who had (not) taken a refresher course. With regard to taking a first-aid course, participants who had done so had significantly better objective knowledge (*M* = 0.67, *t* = 2.82, *p* = 0.005) than those who never had first-aid training (*M* = 0.62). Additionally, they also scored significantly higher on basic first-aid knowledge (*M* = 0.55 versus *M* = 0.47; *t* = 3.01, *p* = 0.002). Regarding subjective knowledge, those who had attended a first-aid course believed that they knew more about first aid than those who had never attended a course (*M* = 3.51 vs. *M* = 2.89, *t* = 6.65, *p* = 0.001). When focusing on those respondents who had attended a refresher course, we found that they had significantly better objective knowledge overall (respectively 0.70 vs. 0.64, *t* = 4.05, *p* < 0.001) and significantly higher scores for basic first-aid knowledge (0.60 vs. 0.51, *t* = 4.09, *p* < 0.001), but they did not have significantly higher scores for the life-threatening cases. Those who had completed a refresher course also reported greater subjective first-aid knowledge than those who did not (*M* = 3.88 vs. 3.28, *t* = 7.72, *p* < 0.001).

The model presented in Table 4 (based on Figure 1) represented a good fit with the data, with a root-mean-square error of approximation value well below the cutoff value of 0.06, and a goodness-of-fit value above the cutoff of 0.95. The value of the comparative fit index was slightly below the recommended cutoff. 

Our model indicates that having completed first-aid training was positively linked to greater basic first-aid knowledge (b = 0.16, *p* < 0.05) and greater subjective knowledge (b = 0.31, *p* < 0.001). We also found that teachers with more years of experience tended to have better basic first-aid knowledge as well (b = 0.21, *p* < 0.01). For life-threatening first-aid knowledge, no statistically significant associations with first-aid predictors were found. 

Regarding subjective first-aid knowledge, our findings showed that having had first-aid training (b = 0.31, *p* < 0.001) and having applied first aid in the past three years (b = 0.26, *p* < 0.001) were linked to more positive views about one’s knowledge of first aid. 

Indirect effects showed that first-aid training (b = 0.10, *p* < 0.001) and having used first aid in the past three years (b = 0.08, *p* < 0.01) were also linked to greater basic first-aid knowledge, through subjective knowledge. No link with life-threatening first-aid knowledge was found.

## 4. Discussion

It is essential that teachers have sufficient knowledge of first aid, as they are likely to encounter accidents at school involving pupils. In these situations, they are responsible for the wellbeing of these children. Despite the importance of first aid, little information exists regarding the objective and subjective knowledge of first aid among teachers. We investigated this among preschool and primary school teachers through an online survey in Flanders, Belgium. Our study, to the best of our knowledge, is the first in Flanders to examine the first-aid knowledge of primary school teachers. We hope that this can be a trigger for more research on this important topic, and for possible policy measures on first aid in schools.

The large majority of respondents (80%) had attended a first-aid course. While no studies on this topic in other European settings were found, studies conducted in other parts of the world show lower rates of first-aid attendance. An Iranian study reported that 40% of respondents had attended first-aid training [7], whereas an Indian study showed that roughly 30% had attended first-aid training [2]. A Turkish study showed that 46% of respondents had received information about first aid or attended a first-aid course [16]. According to an Ethiopian study, 46% of respondents had received a first-aid course [17], while two studies in Saudi Arabia showed that 26% and 30% of respondents had attended a first-aid course, respectively [18,22,23,24,25,26,27]. Taken together, the findings of the present study clearly indicate that a considerably larger share of Belgian teachers have some experience with first-aid courses, compared with select countries in Africa and Asia. 

The average score on overall objective first-aid knowledge was 66% (out of a possible 100), which was similar or somewhat better than in previous studies in Iran (66%) and Saudi Arabia (42%) [7,18], although we should bear in mind that a different survey was used in these studies. In a Slovenian setting, the numbers were worse—for various life-threatening conditions, less than 20% of kindergarten teachers could correctly apply first aid [23]. Nonetheless, it is surprising that the broader dissemination of first aid among teachers in Belgium does not translate into (considerably) better knowledge than in countries where first-aid knowledge is less widespread among teachers. 

The score on life-threatening cases (82%) was higher than the score on basic first-aid cases (53%). However, when taking a closer look at some scores of different questions that make up the life-threatening cases, it should be noted that the scores on the two questions on CPR were far lower than for the other life-threatening cases. For adult CPR, 63% of answers were correct, and for child CPR, only 40% of the answers were correct. This latter result is particularly concerning to report for teachers, given the importance of acting quickly and correctly in emergency situations involving children.

Teachers who had taken a first-aid course achieved higher scores, in line with previous studies [7,18,22,26,27]. Those who had also taken a refresher course scored significantly higher still. Somewhat surprisingly, these higher scores were only found for basic first-aid cases and subjective knowledge. Having completed a first-aid course or refresher course was not linked to higher scores on life-threatening cases. This may signal a lack of focus on life-threatening cases in first-aid training for teachers in Belgium, in line with our conclusions based on the answers for the CPR cases in the previous paragraph.

The role of subjective first-aid knowledge, which has received even less attention in the literature than objective knowledge, cannot be underestimated in the current sample. Our findings indicate that while first-aid training and experience with first aid in the past three years were positively linked to subjective knowledge, they were also partial mediators for objective knowledge. More specifically, having first-aid training and recent experience with applying first aid had a positive indirect effect (through subjective knowledge) on basic first-aid knowledge. This means that having confidence in one’s ability to correctly apply first aid was linked to one’s knowledge about basic first aid cases. A link with life-threatening first-aid cases was not found.

There is little information on how many accidents happen in schools, although we do know that about one in seven accidents involving children happens in a school context [8]. This was even more the case among our sample, where our findings indicated that 66% of teachers had applied some form of first aid in school in the past three years. Thus, teachers appear to use first aid relatively often. It is therefore necessary for teachers to know first aid, even if there is a first aid manager in the school, so that each teacher can at least bridge the time until a first-aid manager is in the classroom—especially in life-threatening circumstances. Although most primary and kindergarten teacher training programs in Belgium include first aid in their curricula, in some colleges this is an optional course [10,11,12,13]. Given that first-aid skills tend to deteriorate over time [14], it is therefore important to encourage not only an initial first-aid course, but also regular refresher courses, so as to contribute to a safer environment in primary schools. More clarity and uniformity should be provided by policymakers in regulations regarding first-aid training and refresher courses for teachers.

There are some limitations that we would like to discuss. It is to be expected that people interested in first aid will be more likely to participate in this type of survey; thus, selection bias is likely to be present in our sample. Additionally, there is no standardized questionnaire for this type of study, although we can observe that other first-aid surveys touch upon similar topics, facilitating comparison across studies. In our study, we also used a survey and not a practical test. Since theoretical knowledge does not always equal practical knowledge, it could be interesting for subsequent studies within this topic to also test teachers’ practical skills in first aid. Future research should also study the potential benefits achieved and experienced in primary school children’s health as a result of teachers’ first-aid knowledge in the school context.

## 5. Conclusions

Our study shows that the first-aid knowledge of primary and preschool teachers in Flanders is higher than in some other countries, but knowledge is still moderate. It also indicates that having completed a first-aid course, as well as completing a refresher course, can positively predict first-aid knowledge. This is why it should be necessary to make first-aid training mandatory in the final stages of teacher training, as many teachers will need first-aid skills during their professional career. Additionally, schools should offer mandatory first-aid refresher courses—for example, during yearly pedagogical study days, which should also involve child CPR.

## Figures and Tables

**Figure 1 children-10-00669-f001:**
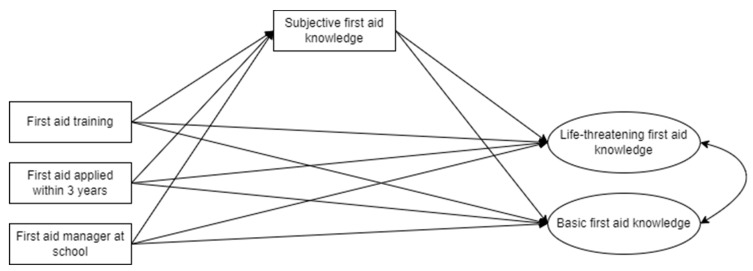
Schematic overview of the links between first-aid indicators, subjective first-aid knowledge, and objective first-aid knowledge.

**Table 1 children-10-00669-t001:** Overview of the study sample (N = 361).

	%
Gender	
Female	94.5
Male	5.4
Years of experience	
<5 years	19.1
5 to 10 years	20.1
11 to 15 years	17.2
16 to 20 years	9.4
>20 years	33.5
First-aid training (yes)	80.3
Refresher course (yes)	38.3
First-aid manager at school (yes)	84.8
Practiced first aid in the past 3 years (yes)	65.7
	Mean score (SD)
Subjective first-aid knowledge (1–5)	3.39 (0.75)
Objective first-aid knowledge (0–1)	
Overall	0.66 (0.13)
Life-threatening cases	0.82 (0.13)
Basic cases	0.53 (0.19)

Note: The question regarding refresher courses was only presented to those who indicated that they had undergone first-aid training. One respondent was removed from the sample due to them indicating “other” on the gender question.

**Table 2 children-10-00669-t002:** Overview of the number and share of correct responses to each case.

Theme	Share of Correct Answers	Number of Correct Answers
Life-threatening cases		
Heat shock	97.8	353
Syncope	97.8	353
Allergic reaction	97.2	351
Choking	97.2	351
Adult CPR	63.2	228
Child CPR	39.6	143
Basic first-aid cases		
First-aid kit	97.0	350
Fall	73.4	265
Burn	66.2	239
Nosebleed	62.3	225
Cuts	46.0	166
Dental injury	41.3	149
Splinter	22.4	81
Bee sting		61

**Table 3 children-10-00669-t003:** Independent-samples *t*-tests on objective and subjective first-aid knowledge.

		Mean	*t*	*p*	Mean	*t*	*p*	Mean	*t*	*p*	Mean	*t*	*p*
First-aid training	Yes	0.67	2.82	0.005	0.82	0.69	0.497	0.55	3.01	0.002	3.51	6.65	0.000
	No	0.62			0.81			0.47			2.89		
Refresher course	Yes	0.70	4.05	0.001	0.84	1.56	0.120	0.60	4.09	0.001	3.88	7.72	0.000
	No	0.64			0.81			0.51			3.28		

**Table 4 children-10-00669-t004:** Standardized effects of predictors on subjective and objective first-aid knowledge.

	Life-Threatening	Basic First Aid	Subjective
Direct effects			
First-aid indicators			
Training	0.78 (0.50)	0.16 * (0.08)	0.31 *** (0.05)
Applied in last 3 years	0.10 (0.12)	0.03 (0.08)	0.26 *** (0.05)
First-aid manager at school	0.19 (0.43)	0.08 (0.08)	−0.04 (0.05)
Subjective knowledge	−0.40 (2.01)	0.31 *** (0.08)	-
Gender (ref.: male)			
Female	0.03 (0.05)	0.07 (0.07)	−0.01 (0.05)
Years of experience	−0.66 (0.65)	0.21 ** (0.07)	0.00 (0.05)
Indirect effects			
First-aid indicators			
Training	−0.13 (0.62)	0.10 *** (0.03)	-
Applied in last 3 years	−0.10 (0.50)	0.08 ** (0.03)	-
First-aid manager at school	0.02 (0.08)	−0.01 (0.02)	-
Fit indices			
RMSEA		0.027	
GFI		0.952	
CFI		0.838	
Chi-squared		182.796	
AIC		314.796	

Note: * *p* < 0.05; ** *p* < 0.01; *** *p* < 0.001. Standardized results reported; standard deviations between brackets. RMSEA = root-mean-square error of approximation; GFI = goodness-of-fit index; CFI = comparative fit index; AIC = Akaike information criterion.

## Data Availability

The data are available from the authors upon reasonable request.

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
