# Peer review of "Teacher Preparedness for Medical Emergencies in Belgian Classrooms: Studying Objective and Subjective First-Aid Knowledge"

_children, 2023, doi:10.3390/children10040669_

Round 1

Reviewer 1 Report

Thank you for the opportunity to review this work. This manuscript is online survey research to evaluate first aid knowledge among teachers in primary education in Flanders, Belgium. Detailed comments about this study are as follows:

-The authors mentioned the related keywords of this manuscript as “objective knowledge; subjective knowledge.” However, those could not be found in the Medical Subject Headings (MeSH) (available from https://meshb.nlm.nih.gov). Providing the other keywords might be more suitable.

-Owing to the non-randomized controlled study of this study, why did the author use the univariable analysis (i.e., t-test) rather than the multivariable regression to adjust the confounding factor, such as the difference in experience among respondents?

-The authors state that "We also controlled for participants' gender and years of experience at teachers." Please provide the detail of how to control these factors.

-If the respondent answered the survey multiple times, whether intention (to receive a higher score than the previous attempt) or non-intention (technical error of the computer or internet), how to manage and ensure the survey's validity as duplicated data record?

-Please provide the response rate of the online survey.

-There was a suspected incomplete column header in Table 3. Please improve the table style.

-In Table 4, please correct the following text ".31*** (.05".

-According to Table 2, why was the child's CPR knowledge, cuts, dental injury, splinter, and bee sting relatively low score, represented by the number of correct answers from the respondents? Was it whether from unclear the question in the survey? Please provide the possible cause to explain this issue in the discussion section.

-In the abstract, the author stated, "Structural equation modelling shows that only previous first aid training, recent experience with first aid, and subjective first aid knowledge were linked to teachers' objective first aid knowledge, particularly for basic first aid." Please explain why subjective first aid knowledge was linked to teachers' objective first aid knowledge in the discussion section.

-Why did question #9 in the survey have 5 choices, while the other questions had only 4 choices?

-In question #12 in the survey, there is a red underline beneath the text inside the figure. Remove it may be suitable for a high-quality figure.

-There were found that the figures of question#12 in the survey in the Appendix might be adapted from the following website: "https://www.wikihow.com/Check-Your-Pulse" and "https://www.wikihow.com/Do-CPR-on-an-Adult" It might be given the reference to these materials.

Author Response

Comment 1: Thank you for the opportunity to review this work. This manuscript is online survey research to evaluate first aid knowledge among teachers in primary education in Flanders, Belgium. Detailed comments about this study are as follows:

The authors mentioned the related keywords of this manuscript as “objective knowledge; subjective knowledge.” However, those could not be found in the Medical Subject Headings (MeSH) (available from https://meshb.nlm.nih.gov). Providing the other keywords might be more suitable.

Response: We have replaced ‘objective knowledge’ and ‘subjective’ knowledge, which can indeed not be retrieved in MeSH, with surveys and questionnaires, which can be found in MeSH.

Comment 2: Owing to the non-randomized controlled study of this study, why did the author use the univariable analysis (i.e., t-test) rather than the multivariable regression to adjust the confounding factor, such as the difference in experience among respondents?

Response: With our analytical strategy, we wanted to provide both univariate analysis through t-tests, as well as multivariate analysis through structural equation modelling (SEM; see Table 4). The latter is a statistical technique that is being increasingly used in all branches of science to represent how various aspects of an observable or theoretical phenomenon are thought to be causally structurally related to one another. Through SEM, we are able to control for confounding factors like experience. In Table 4, you can observe that, with the different types of knowledge as ‘dependent variables (more commonly referred to as endogenous variables in SEM), we are also to add confounders like age and years of experience in a similar fashion to regression analysis, i.e. by incorporating the effect of these variables in addition to more relevant indicators related to first aid.

Comment 3: The authors state that "We also controlled for participants' gender and years of experience at teachers." Please provide the detail of how to control these factors.

Response: See our reply to Comment 2.

Comment 4: If the respondent answered the survey multiple times, whether intention (to receive a higher score than the previous attempt) or non-intention (technical error of the computer or internet), how to manage and ensure the survey's validity as duplicated data record?

Response: Qualtrics, the online tool used to field the survey, allowed us to tick the option to automatically stop individuals from completing the survey twice from the same IP address (which we did). While it is theoretically possible that they completed the survey from different IP addresses, we believe the odds of that to be limited.

Comment 5: Please provide the response rate of the online survey.

Response: As we used a snowball sampling principle (distribution of survey hyperlink via school principals and social media) no response rate of the survey can be determined. It is impossible to find out how many persons received the invitation (via redistributed mail or in their social media feed) and how many persons did indeed participate).

Comment 6: There was a suspected incomplete column header in Table 3. Please improve the table style.

Response: You are correct – this incomplete header has been removed.

Comment 7: In Table 4, please correct the following text ".31*** (.05".

Response: Revised.

Comment 8: According to Table 2, why was the child's CPR knowledge, cuts, dental injury, splinter, and bee sting relatively low score, represented by the number of correct answers from the respondents? Was it whether from unclear the question in the survey? Please provide the possible cause to explain this issue in the discussion section.

Response: While it is difficult to anticipate the potential reasons for these lower scores, we would like to highlight that the knowledge scores in our study are similar to those in other countries. While it is possible that question wording or other factors have contributed to the low scores, a lack of knowledge is of course the most straightforward explanation that we propose is we compare our study to those conducted in other countries. We address and discuss the knowledge scores in our study and compare them to those in other studies (see p. 6):

The average score on overall objective first aid knowledge was 66% (out of a possible 100), which was similar or somewhat better than in previous studies in Iran (66%) and Saudi Arabia (42%) (7,18), although we should bear in mind that a different survey was used in these studies. Nonetheless, it is surprising that the broader dissemination of first aid among teachers in Belgium does not translate into (considerably) better knowledge than in countries where first aid knowledge is less widespread among teachers.

Comment 9: In the abstract, the author stated, "Structural equation modelling shows that only previous first aid training, recent experience with first aid, and subjective first aid knowledge were linked to teachers' objective first aid knowledge, particularly for basic first aid." Please explain why subjective first aid knowledge was linked to teachers' objective first aid knowledge in the discussion section.

Response: Through SEM, we are able to assess if subjective knowledge (partially) mediates the effects of having first aid training. In other words, maybe having had first aid training makes one think they know first aid very well, while in reality this may not be the case – for example, because their training may have been a long time ago. Our results provide some support for a partial mediation of first aid training by subjective knowledge, on basic first aid knowledge.

Comment 10: Why did question #9 in the survey have 5 choices, while the other questions had only 4 choices?

Response: Although we understand why this may seem somewhat strange, the number of answer options did not play a role in the calculation of the knowledge indices. While the number of answer options was relevant to participants, we reduced each item to a correct/incorrect dichotomous variable prior to calculating the knowledge indices. The only indicator that matter was whether or not the correct answer was indicated.

Comment 11: In question #12 in the survey, there is a red underline beneath the text inside the figure. Remove it may be suitable for a high-quality figure.

Response: This has been amended.

Comment 12: There were found that the figures of question#12 in the survey in the Appendix might be adapted from the following website: "https://www.wikihow.com/Check-Your-Pulse" and "https://www.wikihow.com/Do-CPR-on-an-Adult" It might be given the reference to these materials.

Response: We have now credited wikiHow for these images (see Acknowledgements).

Reviewer 2 Report

A very clear and concise paper that examines the role of school teachers in providing first aid to students at primary school levels.  The relevant literature reviewed is comprehensive and draws from a very broad international research and educational publication base.  It is remarkable that the consensus (although this is not systematically discussed) and approval for first aid training for teachers is clearly established.  There is apparently very little debate or discussion about the appropriateness of such training for school teachers or the specific content of such training or its effectiveness.

The article reports the findings of a relatively large online “case based” survey of teachers who are almost all female and the “First aid managers at their school”.  The paper needs to indicate the recruitment requirements to complete the survey and I think that the term “scenario” based has more use and acceptance.  The analyses and findings are clear and should be useful and supportive for those teacher training institutions wanting to move ahead in this area.

The paper is explicitly concerned with the “objective” and “subjective” dimensions of knowledge in this area.  As noted in specific points I believe that it would enhance the paper and its utility if a definition of these differing domains of knowledge were given and why the authors consider the distinction is important in the context of mediating first aid action.

I think that future research to demonstrate the benefits achieved and experienced in primary school children’s health as a result of this knowledge in the school context should be undertaken.

I recommend that be accepted in its current form for publication.

Please note there are some minor points.

Typographical errors

Line 28:  should read:  share of teachers

Difference between 361 surveys [line 102] and analysis sample of 360 [line 143].

Other

Teachers in what school system?  Where are they trained?  Please clarify.

Digital media subject sourcing may have led to a selective sampling interest in the topic note the 85% who were first aid managers at a school.

Although respondents were mostly recruited through social media can they give a comment on why the sample is 95% female and how this might influence the findings.

Author Response

Reviewer 2

A very clear and concise paper that examines the role of school teachers in providing first aid to students at primary school levels.  The relevant literature reviewed is comprehensive and draws from a very broad international research and educational publication base.  It is remarkable that the consensus (although this is not systematically discussed) and approval for first aid training for teachers is clearly established.  There is apparently very little debate or discussion about the appropriateness of such training for school teachers or the specific content of such training or its effectiveness.

The article reports the findings of a relatively large online “case based” survey of teachers who are almost all female and the “First aid managers at their school”.  The paper needs to indicate the recruitment requirements to complete the survey and I think that the term “scenario” based has more use and acceptance.  The analyses and findings are clear and should be useful and supportive for those teacher training institutions wanting to move ahead in this area.

Comment 1: The paper is explicitly concerned with the “objective” and “subjective” dimensions of knowledge in this area.  As noted in specific points I believe that it would enhance the paper and its utility if a definition of these differing domains of knowledge were given and why the authors consider the distinction is important in the context of mediating first aid action.

Response: On page two, we now provide a definition of the types of knowledge that we follow in our study:

Regarding these two types of knowledge, we follow the definition outlined by Han [23]: “Objective knowledge refers to how much an individual actually knows and subjective knowledge (also known as perceived or self-assessed knowledge) involves how much an individual thinks he/she knows […]”.

Comment 2: I think that future research to demonstrate the benefits achieved and experienced in primary school children’s health as a result of this knowledge in the school context should be undertaken.

Response: We fully agree with this sentiment and have added this in the discussion as an avenue for future research (see p. 7).

I recommend that be accepted in its current form for publication.

Please note there are some minor points.

Typographical errors

Comment 3: Line 28:  should read:  share of teachers

Difference between 361 surveys [line 102] and analysis sample of 360 [line 143].

Response: Thank you for noting these oversights. These points have been revised.

Other

Comment 4: Teachers in what school system?  Where are they trained?  Please clarify.

Digital media subject sourcing may have led to a selective sampling interest in the topic note the 85% who were first aid managers at a school.

Although respondents were mostly recruited through social media can they give a comment on why the sample is 95% female and how this might influence the findings.

Response: We provided more detail about the teachers on p. 3.

Regarding your other points: we had made a note on the aselect sampling in the limitations (p. 7):

It is to be expected that people interested in first aid will be more likely to participate in this type of survey, and thus, selection bias is likely to be present in our sample.

We believe that there are two reasons why the sample is 95% female. First, the gender distribution of teachers in Flanders is highly uneven: about 70% are women and 30% are men. Additionally, the methodology used for data collection (online survey research) is also known to yield a higher participation rate of women compared to men. These two combined factors likely contributed to the uneven gender distribution in our sample.

Reviewer 3 Report

Interesting article, I think it is suitable for publication after some minor changes

Here are some comments.

1. you should consider changing the keywords to better fit the subject matter covered

2. share of correct answers should be put differently in Table 2, it seems that it would be better to give the percentage first 

3. conclusions should be shortened by limiting only to the information obtained from the survey.

4. consider increasing the number of cited articles. This is quite a widely analyzed topic and 23 references are far too few considering the scope of the conclusions drawn.

5. The limitation section should be expanded to take into account the limitations of the survey conducted, including the lack of detailed information on participants' knowledge of first aid, the lack of verification of information provided in the form of the survey 

Author Response

Interesting article, I think it is suitable for publication after some minor changes

Here are some comments.

Comment 1: you should consider changing the keywords to better fit the subject matter covered

Response: In line with Comment 1 from Reviewer 1, we have made several revisions to the keywords. We have replaced ‘objective knowledge’ and ‘subjective’ knowledge, which can indeed not be retrieved in MeSH, with surveys and questionnaires, which can be found in MeSH.

Comment 2: share of correct answers should be put differently in Table 2, it seems that it would be better to give the percentage first

Response: This has been changed.

Comment 3: conclusions should be shortened by limiting only to the information obtained from the survey.

Response: We believe it is customary to end a conclusion with a take-away message for readers. The first half of the conclusion is indeed a brief summary of our results, while in the second half, we try to provide some kind of recommendations to policy makers, practitioners… about how first aid for teachers should (in our view, based on the results of the study) be addressed in the future.

Comment 4: Consider increasing the number of cited articles. This is quite a widely analyzed topic and 23 references are far too few considering the scope of the conclusions drawn.

Response: You are quite right – we have added various extra references to strengthen some of the conclusions drawn.

Comment 5: The limitation section should be expanded to take into account the limitations of the survey conducted, including the lack of detailed information on participants' knowledge of first aid, the lack of verification of information provided in the form of the survey 

Response: We have expanded our limitations section to add some elements that you highlighted. We are unsure about adding the lack of detailed information on participants’ knowledge of first aid as a limitation, given that this is precisely what the study is about. However, although we did not mention this in the article, we did provide the correct answers to the cases to respondents when the survey ended. This way, these answers couldn’t affect their response, but they were still informed of the correct action to take in each case. Given that this wasn’t relevant for completing the survey, we did not report it.

Reviewer 4 Report

The present work addresses an extremely current and important topic in the school education of young people regarding the provision of first aid.
The quality of the work is a good one, but I have a few remarks, the maximum time period between refresher courses is important and should be mentioned in the conclusions.
Also, the authors should mention when during the school period these courses should be carried out by the students, and at what time intervals they should be repeated.

Author Response

Comment 1: The present work addresses an extremely current and important topic in the school education of young people regarding the provision of first aid.

The quality of the work is a good one, but I have a few remarks, the maximum time period between refresher courses is important and should be mentioned in the conclusions.

Also, the authors should mention when during the school period these courses should be carried out by the students, and at what time intervals they should be repeated.      
Response: We have added information regarding the time period between refresher courses (yearly), and when the courses should be taken by students in their training (in the final stages, as close to their professional careers as possible) (see p. 7).

Reviewer 5 Report

I thank the authors for the opportunity to read this interesting manuscript.

I have some doubts about the methodology used in the study.

Have the authors calculated the sample size needed to carry out the study? It should indicate the calculation that the authors made on the necessary sample size and detail the data thereof (power, significance,...).

The number of subjects targeted by the study should be indicated, ie the teachers in Flanders who could have participated in the study.

The variables that the authors analyze are percentages, however, they artificially transform them into quantitative variables. The new variable created 0-1 in an artificial transformation of a percentage. The authors must carry out the calculations with the real percentages of correct answers and with the statistical methods for these variables and not for quantitative variables.

The same happens with the ordinal categorical variable measured from 1-5. Stockings cannot be done. It is not a continuous quantitative variable and student's t cannot be calculated. The authors should redo the calculations treating it as an ordinal qualitative variable.

Author Response

I thank the authors for the opportunity to read this interesting manuscript.

I have some doubts about the methodology used in the study.

Comment 1: Have the authors calculated the sample size needed to carry out the study? It should indicate the calculation that the authors made on the necessary sample size and detail the data thereof (power, significance,...).

The number of subjects targeted by the study should be indicated, ie the teachers in Flanders who could have participated in the study.

Response: The total number of subjects targeted (82,520 teachers) has been added in the study. A pre-study power analysis indicated that, in order to attain a 95% confidence level and 5% margin of error on a population size of 82,520 and a population proportion of 2.5% (which is double the actual proportion, but we wanted to be stringent in our case selection), we required 38 respondents. With the 361 respondents included in the study, we far exceeded this number.

Comment 2: The variables that the authors analyze are percentages, however, they artificially transform them into quantitative variables. The new variable created 0-1 in an artificial transformation of a percentage. The authors must carry out the calculations with the real percentages of correct answers and with the statistical methods for these variables and not for quantitative variables.

The same happens with the ordinal categorical variable measured from 1-5. Stockings cannot be done. It is not a continuous quantitative variable and student's t cannot be calculated. The authors should redo the calculations treating it as an ordinal qualitative variable.

Response: Creating composite scores based on a number of categorical variables is a common practice in various branches of science (e.g. clinical psychology). The percentages presented cannot be used for calculation given that they are aggregates across all respondents, while at the respondent level, the score is 0-1 (correct or incorrect). It would be possible to calculate the % of correct answers per respondent, but then you would simply have the 0-1 distribution that we had now * 100, which would yield the same results. After all, to calculate the 0-1 score took the number of correct answers (for example: 8 out of 14), and divided them, providing an index for the current example of 0.57 (or 57% correct answers). So, in a way, the index represents the share of correct answers per person.

Regarding the categorical variable measured 1-5, there is some discussion in the literature. You are quite correct that we have treated it as a continuous quantitative variable. While some may argue that it is indeed an ordinal categorical variable, other scholars have clearly shown that these types of Likert scales can be treated as continuous (Johnson & Creech, 1983; Norman, 2010; Sullivan & Artino, 2013; Zumbo & Zimmerman, 1993). None the less, the discussion regarding the appropriate use of these types of Likert scales remains a hot topic among survey methodologists.

Johnson, D.R., & Creech, J.C. (1983). Ordinal measures in multiple indicator models: A simulation study of categorization error. American Sociological Review, 48, 398-407.

Norman, G. (2010). Likert scales, levels of measurement and the “laws” of statistics. Advances in Health Sciences Education, 15(5), pp. 625-632. Retrieved from: https://link.springer.com/article/10.1007%2Fs10459-010-9222-y#citeas.

Sullivan, G. & Artino Jr., A. R. (2013). Analyzing and Interpreting Data From Likert-Type Scales. Journal of Graduate Medical Education. 5(4), pp. 541-542.

Zumbo, B. D., & Zimmerman, D. W. (1993). Is the selection of statistical methods governed by level of measurement? Canadian Psychology, 34, 390-400.

Round 2

Reviewer 1 Report

The authors have modified the text following the reviewers' suggestions. I think the manuscript deserves publication.

Reviewer 5 Report

I thank the authors for the response to the comments made by the reviewer.